# Yeast Associated with Rice Phylloplane and Their Contribution to Control of Rice Sheath Blight Disease

**DOI:** 10.3390/microorganisms8030362

**Published:** 2020-03-03

**Authors:** Parichat Into, Pannida Khunnamwong, Sasitorn Jindamoragot, Somjit Am-in, Wanwilai Intanoo, Savitree Limtong

**Affiliations:** 1Department of Microbiology, Faculty of Science, Kasetsart University, Bangkok 10900, Thailand; parichat_into@hotmail.com (P.I.); pannida_minn@hotmail.com (P.K.); 2National Center for Genetic Engineering and Biotechnology (BIOTEC), National Science and Technology, Development Agency (NSTDA), Pathum Thani 12120, Thailand; sasitorn@biotec.or.th (S.J.); somjit@biotec.or.th (S.A.-i.); 3Department of Plant Pathology, Faculty of Agriculture, Kasetsart University, Kamphaeng Saen Campus, Nakhon Pathom 73140, Thailand; wanwilai.i@ku.th; 4Academy of Science, The Royal Society of Thailand, Bangkok 10300, Thailand

**Keywords:** phylloplane yeast, antagonistic yeast, rice pathogenic fungi, rice sheath blight, biological control

## Abstract

The phylloplane is an important habitat for yeasts and these yeasts may have antagonistic activities against pathogens and could be used as biocontrol agents. To investigate rice phylloplane yeasts, 282 strains were isolated from 89 rice leaf samples and identified as 15 known yeast species in the phylum Ascomycota and 35 known and two potential new species in the phylum Basidiomycota. The majority of rice phylloplane yeasts belonged to the phylum Basidiomycota. The evaluation of antagonistic activities of 83 yeast strains against rice pathogenic fungi *Pyricularia oryzae*, *Rhizoctonia solani*, *Fusarium moniliforme*, *Helminthosporium oryzae* and *Curvularia lunata* revealed that 14 strains inhibited these pathogens. Among the antagonistic strains, *Torulaspora indica* DMKU-RP31, *T. indica* DMKU-RP35 and *Wickerhamomyces anomalus* DMKU-RP25 inhibited all rice pathogens tested, and the production of volatile organic compounds, fungal cell wall degrading enzymes and biofilm were the possible antagonistic mechanisms against all rice pathogens tested in vitro. These yeast strains were evaluated for controlling rice sheath blight caused by *R. solani* in rice plants in the greenhouse and were found to suppress the disease by 60.0–70.3%, whereas 3% validamycin suppressed by 83.8%. Therefore, they have potential for being developed to be used as biocontrol agents for rice sheath blight.

## 1. Introduction

The term phylloplane refers to the parts of plants above ground and dominated by leaves and is an important habitat for microorganisms [1]. Growth of phylloplane microorganisms depends on organic compounds secreted from the plant itself and organic compounds from external sources [2,3]. The phylloplane of plants in both temperate and tropical regions has been reported to be colonized by yeasts belonging to both phyla, Ascomycota and Basidiomycota, although the majority of the strains are in Basidiomycota [4,5,6,7,8].

Rice is one of the most widely produced and consumed staple foods in the world, especially in Asia [9]. In Thailand the rice cultivation area in the year 2017/2018 was approximately 59.2 million hectares, with a production of 24.9 million tons (http://www.oae.go.th/view/1/). The rice species cultivated in Thailand is commonly known as Asian rice (*Oryza sativa* L.). One of the major causes of decreases in rice production is diseases caused by pathogenic fungi. The major rice diseases caused by fungal pathogens in Thailand are blast (caused by *P. oryzae*), sheath blight (caused by *R. solani*), bakanae (caused by *F. moniliforme*), brown spot (caused by *H. oryzae*) and dirty panicle (caused by *Cu. lunata* and *H. oryzae*) (Rice Department, Ministry of Agriculture and Cooperatives of Thailand, 2014). Rice sheath blight disease causes yield losses of 25%–35% of Thai rice production [10]. This disease is the second most important rice disease worldwide [11].

The management of rice diseases caused by fungi is mainly based on the use of the chemical fungicides such as Carbendazim^®^, Validamycin^®^, Propiconazole^®^ and Mancozeb^®^ [12,13,14]. However, the use of chemical fungicides is not a long-term solution and is becoming less acceptable due to increasing residues, toxicity to non-target organisms and other health and environmental hazards [15]. Biological control is an environmentally friendly alternative approach to plant disease management. In the last two decades, biological control based on using antagonistic yeast has been demonstrated. Antagonistic yeasts have been sought for use as biological control agents for plant and post-harvest diseases. For example, *Papiliotrema* (*Cryptococcus*) *flavescens* and *Sporobolomyces roseus*, reduced lesion density and necrosis of red stalk rot of maize caused by *Colletotrichum graminicola* when applied as a mixture to maize plants [16], *Saccharomyces cerevisiae*, *Candida albicans* and *Candida sake* were reported to significantly reduce the powdery mildew and Cercospora leaf spot diseases on sugar beet [17] and *Rhodotorula* (*Rhodosporidium*) *kratochvilovae* and *Papiliotrema* (*Cryptococcus*) *laurentii* UM108 suppressed the powdery mildew disease on wheat [18].

Antagonistic yeasts use both direct and indirect mechanisms to inhibit plant pathogens, including production of volatile organic compounds, cell wall degrading enzymes, siderophores and biofilm as well as competition of nutrients, and phosphate and zinc oxide solubilization [18,19,20,21,22,23,24,25]. Volatile organic compounds (VOCs) are substances with a low molecular weight (lower than 300 Da), low polarity and high vapor pressure [26]. Emission of VOCs by antagonistic yeasts have proven to be one of the important direct antagonistic mechanisms against pathogenic fungi. *Candida intermedia* was found to produce VOCs that inhibited mycelium growth of *Botrytis cinerea* [27]. *Candida maltosa* NP9 emitted VOCs that inhibited spore germination of *Aspergillus brasiliensis* [28]. *Sporidiobolus pararoseus* was found to produce VOCs that effectively inhibited both the conidial germination and the mycelial growth of *B*. *cinerea* [29]. Hua et al. [30] reported that VOCs produced by *Wickerhamomyces anomalus* WRL-076 inhibited growth and aflatoxin production of *Aspergillus flavus*. Secretion of fungal cell wall degrading enzymes, especially β- 1, 3-glucanase and chitinase, by antagonistic yeasts is one of the direct antagonistic activities against pathogenic fungi. These enzymes hydrolyze polymeric compounds in the fungal cell wall, and this directly suppresses activities and/or induces death. *Candida oleophila*, *Meyerozyma guilliermondii* and *Pichia membranifaciens* were reported to produce β- 1, 3-glucanase and chitinase, which kill pathogenic fungi [22]. *M. guilliermondii* produced β-1,3-glucanases and chitinase, and higher production of both enzymes was found when cultivated in a medium supplemented with cell wall fragments of *Colletotrichum capsici* [31]. *P. membranifaciens* suppressed the growth of *B. cinerea* through the production of β-1,3-glucanases [32]. *Meyerozyma caribbica* showed a high antagonistic potential against *Colletotrichum gloeosporioides* in mango through the production of hydrolytic enzymes such as chitinase and β-1,3-glucanase [23]. Siderophores are secondary metabolites which are low molecular weight compounds with a high affinity for iron. When siderophore producing antagonistic microorganisms are applied in the agricultural field, they suppress the pathogens’ growth or reduce their metabolisms by competition for iron, resulting in a decrease in the pathogens [33]. *Rhodotorula glutinis* has been shown to produce rhodotorulic acid, a hydroxamate type siderophore, which suppresses various plant pathogenic fungi [34]. This antagonistic yeast species was also reported to reduce *B. cineria* spore germination and disease caused by this pathogenic fungus in biocontrol experiments on apple fruit [35]. Biofilm is a group of microbial cells embedded within a matrix of extracellular polymeric substance produced by the cells, which adhere to a surface. Biofilm formation on a plant will protect the plant from destruction by pathogens and is related to the competition for nutrients on the surface of the plant [23,36]. In addition, yeasts cells in biofilm can destroy fungal pathogens by secretion of fungal cell-wall degrading enzymes during the adhesion process [36]. Competition for nutrients between antagonists and pathogenic fungi is among the direct antagonistic mechanisms. This mechanism is based on the ability of antagonistic yeasts to rapidly colonize and multiply on the plant surface and subsequently to compete with the pathogens for nutrients and space [24,31,37]. Reduced efficiency of *W. anomalus* and *S. cerevisiae* against *Pichia digitatum* in orange was obtained from this mechanism [38]. Tian et al. [39] reported that *Metschnikowia pulcherrima* could compete for nutrients under in vivo condition when glucose was added to wounds on mango fruits. Some microorganisms are capable of solubilizing insoluble essential elements such as phosphate and zinc oxide in the environment to their soluble forms, which are subsequently uptaken by plants [21].

This work aimed to study yeasts in the phylloplane of rice, to evaluate the antagonistic activities and mechanisms against rice pathogenic fungi of the rice phylloplane yeasts and to evaluate the efficacy of the selected antagonistic yeasts for controlling the rice sheath blight disease in rice plants in a greenhouse.

## 2. Materials and Methods

### 2.1. Rice Leaf Collection and Phylloplane Yeast Isolation

Eighty-nine samples of green and undamaged leaves of rice were collected at random from rice fields in nine provinces in Thailand between December 2011 and March 2012 (Table 1). Each sample was a composite sample consisting of many leaves collected from different rice plants in a small area. Each sample was put in a plastic bag and transferred to the laboratory in an ice box. The samples were stored at 8 °C and yeast isolation was initiated within 5 days.

Yeast were isolated from the rice phylloplane by a plating of leaf washings as described by Surussawadee et al. [40]. Three grams of leaf from all the collected leaves in each sample were suspended in 50 mL of 0.85% sodium chloride solution in a 250 mL Erlenmeyer flask and shaken on a rotary shaker at room temperature (27 ± 3 °C) and 150 rpm for 1 h. An aliquot of 100 µL of the washing solution was then spread on yeast extract malt extract (YM) agar (3 g/L yeast extract, 3 g/L malt extract, 5 g/L peptone, 10 g/L glucose and 20 g/L agar), supplemented with 250 mg/L sodium propionate and 200 mg/L chloramphenicol and incubated at 25 °C until yeast colonies appeared. The colonies of different morphologies were selected and purified by streaking on YM agar. The purified yeast strains were suspended in YM broth supplemented with 150 g/L glycerol and maintained at 80 °C.

### 2.2. Yeast Identification

Yeasts were identified by analysis of the D1/D2 region of the large subunit (LSU) rRNA gene sequence similarity, and in some instances, the internal transcribed spacer (ITS) region sequence similarity was also analyzed. The sequence of the D1/D2 region of the LSU rRNA gene and the ITS region were determined from PCR products amplified from genomic DNA extracted from yeast cells following the methods previously described by Limtong et al. [41]. Amplification of the D1/D2 of the LSU rRNA gene was carried out using PCR with the forward primer NL1 (5′-GCATATCAATAAGCG GAGGAAAAG-3′) and the reverse primer NL4 (5′-GGTCCGTGTTTCAAGACGG-3′) [42]. The ITS region was amplified with the forward primer ITS1 (5′-TCCGTAGGTGAACCTGCGG-3′) and the reverse primer ITS4 (5′-TCCTCCGCTTATTGATATGC-3′). The PCR products were checked using agarose gel electrophoresis and purified by using the HiYield Gel/PCR Fragment Extraction Kit (RBC Bioscience). The purified products were sequenced commercially by Macrogen (Geumcheon -gu, Seoul, Korea) using the primers NL1 and NL4 for the D1/D2 region and the primers ITS1 and ITS4 for the ITS region. The sequences were compared pairwise using a BLAST search [43]. For identification of ascomycetous yeasts, strains showing greater than 1% nucleotide substitutions in the D1/D2 region were considered to be different species and strains with 0–3 nucleotide differences were treated as conspecific species [42]. For basidiomycetous yeast identification, strains differing by two or more nucleotide substitutions were considered to represent different species [44]. When possible, complete ITS sequences were also analyzed in order to confirm D1/D2-based identifications. When necessary, the “potential new species” designation was used.

Phylogenetic analysis based on the sequences of the D1/D2 region of the LSU rRNA gene was used for confirming yeast identification with pairwise sequence similarity analysis. The nucleotide sequences of the type strains of the related species were obtained from NCBI (www.ncbi.nih.gov) databases. The sequences of yeast strains were aligned with their related species using MUSCLE [45] provided with the MEGA version 7.0 software package. A phylogenetic tree was constructed from the evolutionary distance data using the general time reversible (GTR) model and the maximum likelihood analysis performed with MEGA 7.0 [46] The confidence level of the clade was estimated using bootstrap analysis (1000 replicates). Yeast sequence data have been submitted to the GenBank database.

### 2.3. Selection of Antagonistic Yeasts Capable of Antagonize Fungi Cause Rice Diseases

The antagonistic activities of 83 strains of rice phylloplane yeast against five rice pathogenic fungi (Table 2) were determined by dual cultivation of yeast and pathogenic fungi on a potato dextrose agar (PDA; 4.0 g/L potato, infusion from (solids), 20.0 g/L glucose and 15.0 g/L agar-agar) dish following the method of Rosa et al. [21] with slight modification. The rice pathogenic fungus and yeast were grown side by side on PDA agar in a Petri dish. A loop of an active yeast culture (24 h at 25°C on yeast extract malt extract (YM) agar) was inoculated near the edge of the dish and then incubated at 25 °C for 2 days. A 5 mm diameter disk of actively growing fungal mycelium from a 7-day-old culture on PDA was cut with a cork borer and inoculated at the opposite edge. A PDA dish inoculated with only the pathogenic fungi was used as a control. Dishes inoculated with *R. solani* were incubated at 25 °C for 3 days, while those inoculated with the other plant pathogenic fungi were incubated at 25 °C for 7 days. Three replications were performed. The pathogenic fungal growth reduction was determined by measuring the radius of the fungal colony cultured with yeast compared with the control.

Inhibition of fungal growth (%) = [(R1-R2)/R1] × 100.

R1 = radius of fungal colony cultured alone; R2 = radius of fungal colony cultured with yeast.

### 2.4. Determination of Antagonistic Mechanisms of Antagonistic Yeasts In Vitro

#### 2.4.1. Production of Antifungal Volatile Organic Compounds

Production of VOCs against plant pathogenic fungi was measured using the double dishes assay technique as described by Di Francesco et al. [47] with slight modification. An aliquot (100 µL) of the antagonist yeast cell suspension of 10^8^ cells/ mL was spread on a PDA dish. After 48 h, the cover of the dish was removed and the bottom with yeast inoculation was inverted and placed upside down on the bottom of a PDA dish inoculated with a fungal mycelial plug (5 mm in diameter). The two bottom dishes were sealed with a double layer of parafilm and incubated at 25 °C for 7 days. The control was a PDA dish inoculated with only the pathogenic fungus. Three replications were performed. The fungal growth inhibition was calculated with the flowing formula:

Inhibition of mycelial growth (%) = [(D1-D2)/D1] × 100.

D1 = diameters of fungal colony cultured alone.

D2 = diameter of fungal colony cultured with yeast.

#### 2.4.2. Production of β-Glucanase and Chitinase

The production of the two fungal cell wall lytic enzymes, β-glucanase and chitinase, by the antagonistic yeasts was determined by cultivation in potato dextrose broth (PDB). A loop of active yeast cells (24 h at 25 °C on YM agar) was inoculated to 50 mL PDB in a 250 mL Erlenmeyer flask and incubated on a rotary shaker at 150 rpm for 5 days. The culture broth was collected, and cells were separated by centrifuging at 10,000 *g* for 5 min. The supernatant was analyzed for β-glucanase and chitinase activity. Three replications were performed.

To determine β -1, 3-glucanase activity, the colorimetric quantification of glucose (reducing sugar) released from laminarin was used. The enzymatic reaction was performed by mixing 200 µL of the cell free culture broth, 50 µL of sodium acetate buffer (0.1 M, pH 5.0) and 250 µL of laminarin (4 mg/mL), incubated at 37 °C for 1 h, and the reducing sugar concentration was determined using the method of Miller [48]. Briefly, 250 µL of dinitrosalicylic acid reagent was added to 250 µL of the enzymatic reaction mixture and incubated at 100 °C for 5 min, and the absorption was measured with a spectrophotometer at 540 nm. The enzymatic activity was expressed in U/mL, in which one unit of activity (U) was defined as 1 µg of reducing sugars released from laminarin per minute under the assay conditions.

Chitinase activity was evaluated by estimating the release of N-acetyl glucosamine (NAG) from the substrate colloidal chitin. A volume of 100 µL of the cell free culture broth was mixed with 200 µL of Mcllvaine buffer (50 mM citric acid, 100 mM sodium phosphate, pH 6.0) and 100 µL of 0.01% w/v colloidal chitin (prepared in Mcllvaine buffer). After incubation at 37 °C for 15 min, the reducing sugar concentration was determined using the method of Miller [48], the same as used to determine the reducing sugar concentration in investigation of b-1, 3-glucanase activity, except that the absorption was measured at 575 nm. The enzymatic activity was expressed in U/mL, in which one unit of activity (U) was defined as 1 µg of reducing sugars released from colloidal chitin per minute under the assay conditions.

Colloidal chitin was prepared as described by Khan et al. [49]. Chitin powder (40 g) was dissolved in 500 mL of concentrated hydrochloric acid and continuously stirred at 4°C for 1 h. The hydrolyzed chitin was washed several times with distilled water to completely remove acid and hence bring it to pH 6–7. Then the colloidal chitin was filtered through Whatman filter paper No.1. The colloidal chitin was collected, frozen in liquid nitrogen for lyophilization and stored at −20 °C until use.

#### 2.4.3. Competition of Nutrients

To determine the effect of nutrient concentration on the pathogenic fungal mycelia growth reduction by the antagonistic yeast, the method of Zhang et al. [37] with slight modification was used. The dual cultivation of yeast and the pathogenic fungi was performed, the same as described previously except that PDA with different nutrient concentrations, namely, standard nutrient concentration (39 g/L PDA powder), half of the standard nutrient concentration (19.5 g/L PDA powder), one-fourth of the standard of nutrient concentration (9.7 g/L PDA powder), and one-tenth of the standard nutrient concentration (3.9 g/L PDA powder) were used. Three replications were performed. The radius of pathogenic fungal colony cultivated alone and that of a fungal colony cultivated with yeast was determined after 3 days at 25 °C. Inhibition of fungal growth was calculated as described previously.

#### 2.4.4. Phosphate and Zinc Oxide Solubilization

To determine phosphate and zinc oxide solubilization, Pikovskaya’s agar [50] and zinc oxide agar [51] were used. A yeast cell suspension was prepared by mixing a 2-day-old culture grown on YM agar with sterile normal saline and adjusting the suspension to an optical density at 600 nm (OD_600_) of 0.10. The yeast cell suspension (5 µL) was dropped onto the surface of Pikovskaya’s agar or zinc oxide agar in a dish and incubated at 25 °C for 5 days. The halo zone diameter and the colony diameter were then measured. Three replications were performed. The phosphate or zinc oxide solubilization efficiency (SE) was calculated as a ratio of the halo zone diameter and the colony diameter.

#### 2.4.5. Siderophore Production

Investigation of siderophore production by the selected antagonistic yeast was carried out by cultivation on chrome azurol S (CAS) blue agar in a Petri dish [52] A yeast cell suspension was prepared by mixing a loop full of a 2-day-old culture grown on YM agar in 3 mL of 0.85% sterile normal saline and adjusting the suspension to an optical density at 600 nm (OD_600_) of 0.10. The yeast cell suspension (10 µl) was dropped onto the surface of CAS blue agar and incubated at 25 °C for 10 days, in the dark. Three replications were performed. The medium color change (from blue to purple or yellow) around the colony indicated siderophore production.

#### 2.4.6. Biofilm Formation

The antagonistic yeast strains were assessed for biofilm formation using the method described by Růžička et al. [53] with slight modification. A yeast cell suspension was prepared by mixing a 2-day-old culture grown on PDA with sterile water to reach an optical density at 600 nm (OD_600_) of 0.5. The yeast suspension (20 µL) was inoculated into each well of a 96-well microtiter plate containing 180 µL PDB, and the microtiter plate was incubated for 48 h at 25 °C. The negative control was a well containing only PDB. Three replications were performed. After incubation, the wells were emptied, rinsed with water and air-dried at room temperature. The adherent biofilm layer was stained with an aqueous solution of 1% (*w*/*v*) violet crystal for 20 min, rinsed with water and air-dried. The stained biofilm layer was eluted from each well with 200 µL of ethanol, and the absorbance (A) of each well was measured at 620 nm. Biofilm formation was considered as positive in a well where mean A of the treatment was higher than the mean A of the negative control.

### 2.5. Controlling of Rice Sheath Blight Disease in Rice Plants in the Greenhouse by the Selected Antagonistic Yeasts

The antagonistic yeast strains that strongly inhibited mycelium growth of *R. solani* DOAC 1406 in dual cultivation were selected for determination of their ability to control rice sheath blight disease in rice plants in the greenhouse.

An antagonistic yeast cell suspension was prepared using the method of Spadaro et al. [54] with slight modification. The selected yeast strain was cultured in 50 mL of yeast extract peptone dextrose (YPD) broth (10 g/L yeast extract, 20 g/L peptone and 20 g/L glucose) in a 250 mL Erlenmeyer flask and incubated on a rotary shaker at 150 rpm and 25 °C for 24 h. Cells were collected using centrifugation at 5000 *g* for 10 min, re-suspended in sterile Ringer’s solution and quantified with a haemacytometer to reach a concentration of 10^8^ cells/mL.

Pathogenic fungus was cultivated on sterile rice straw. The rice straw was cut into pieces (1 × 2 cm), 15 g of the cut rice straw was put in a clean plastic bag (20 × 30 cm), and 15 mL distilled water was added to make the moisture content approximately 50%. The open end of the plastic bag was narrowed and closed with a cotton plug to make a closed system. The rice straw in the plastic bag was sterilized using autoclave at 121.5 °C for 15 min. After cooling, *R. solani* was inoculated by using 3 plugs (0.5 mm diameter) of a 3-day-old culture grown on PDA cut from the edge of a colony and incubated at 28 ± 2 °C for 14 days. The infested rice straw (5 g) was put on rice straw paper (20 × 7 cm) before use.

A greenhouse experiment was conducted in rice plants grown in pots using complete randomized design (CRD) with five replications. Chai Nat 1 rice cultivar, which is susceptible to rice sheath blight disease, was used. Rice seeds were sterilized in 10% Clorox solution for 1 min, rinsed three times with sterile distilled water and soaked in sterile distilled water for 24 h for imbibition prior to the germination trial [9]. Sterile seeds were grown in sterile loam soil in a plastic nursery basket for 20 days in a greenhouse and watered daily. One 20-day-old rice seedling was transplanted into a pot (22.5 cm diameter and 22 cm height) containing 5 kg of sterile loam soil. Tap water was added to the pots such that the water level was 3 cm above the soil surface, and that water level was kept constant by adding water daily. Twenty mL of a selected antagonistic yeast cell suspension was sprayed on the sheath of the rice plant. One hour after yeast inoculation, the rice straw paper containing the infested rice straw (5 g) was inoculated at 5 cm above water level on the sheath of a rice plant (45 days old) [55]. Five days after pathogenic fungus inoculation, the rice straw paper was removed. The fungicide, 3% (*w*/*v*) validamycin, was used for comparison. A rice plant inoculated with pathogenic fungi without any treatment was used as a positive control. Spraying of all treatments was repeated at 5 and 10 days after pathogenic fungus inoculation. Five replications were performed. The sheath blight disease symptoms were observed, and lesion height was measured 15 days after pathogenic fungus inoculation. The disease incidence and disease suppression of the treatment and control were calculated using the following formulae [56]:

Sheath blight disease incidence (%)

= (Average sheath blight lesion height/Average plant height) × 100.

Sheath blight disease suppression (%)

= [(Incidence of positive control-Incidence of treatment)/ Incidence of positive control)] × 100.

### 2.6. Yeast Population and Development of Sheath Blight Lesion on Rice Plants

To examine yeast population and development of sheath blight lesion on rice plants during controlling rice sheath blight in the greenhouse experiment, *T. indica* DMKU-RP31 was used. The experiment was conducted by inoculation of the selected antagonistic yeast and *R. solani* DOAC 1406, as described in Section 2.5. Yeast population and disease lesion development on rice sheath 3, 4 and 5 day(s) after the first spraying, 0 (1 h), 2, 4 and 5 day(s) after the second spraying, and 0 (1 h), 2, and 5 day(s) after third spraying were investigated. Three replicates were performed.

The yeast population on a rice sheath was determined using a swab test. The predetermined area of rice sheath surface was swapped using the sterile cotton swab. The cotton swab was then placed into a test tube containing sterile Ringer’s solution and shaken for 30–45 s to remove yeast cells from the cotton swab. Enumeration of the yeast population was carried out using the method of Nix et al. [57] with slight modifications. The washing solution was spread onto the surface of YM agar supplemented with 250 mg/L sodium propionate and 200 mg/L chloramphenicol. The inoculated plate was incubated at 25 °C for 48 h and the yeast colonies were counted. The population of yeast was expressed as colony forming units (CFU) per square centimeter (cm^2^) of rice sheath surface. The development of sheath blight lesion was measured as the lesion height.

### 2.7. Statistical Analysis

The data were subjected to analyses of variance (ANOVA) using IBM^®^ SPSS statistics software version 22 (Armonk, New York, United States) for Windows. Statistical significance was evaluated using Duncan’s multiple range test (DMRT) and the significance level of *p* < 0.05 was considered as being significantly different.

## 3. Results

### 3.1. Rice Phylloplane Yeast Isolation and Identification

From 89 rice leaf samples collected from nine provinces in Thailand, 282 yeast strains, each representing a different morphology in individual sample, were obtained (Table 1). All yeast strains were identified by sequence similarity and phylogenetic analysis of the D1/D2 region of the LSU rRNA gene and, in some instances, by the ITS region as well. Forty-four strains were identified to be 15 known yeast species belonging to the subphylum Saccharomycotina, phylum Ascomycota (Table 3, Appendix A, Appendix A). The known ascomycetous yeast species belonged to nine genera: *Blastobotrys* (one species), *Candida* (five species), *Debaryomyces* (one species), *Hyphopichia* (one species), *Kodamaea* (one species), *Meyerozyma* (two species), *Torulaspora* (one species), *Wickerhamomyces* (two species) and *Yamadazyma* (one species). A total of 238 strains were found to belong to the phylum Basidiomycota consisting of 15 known and one potential new species in the subphylum Agaricomycotina, 11 known and a potential new species in the subphylum Pucciniomycotina and nine known species in the subphylum Ustilaginiomycotina (Table 3, Appendix A, Appendix A). Among the strains belonging to the phylum Basidiomycota, 229 strains were identified to be 35 known species of yeasts and yeast-like fungi in 16 genera. These were *Hannaella* (four species), *Papiliotrema* (seven species), *Saitozyma* (one species), *Trichosporon* (three species), *Occultifur* (one species), *Rhodotorula* (four species), *Sakaguchia* (one species), *Sporobolomyces* (four species), *Sporidiobolus* (one species), *Symmetrospora* (one species), *Dirkmeia* (one species), *Jaminaea* (one species), *Kalmanozyma* (one species), *Moesziomyces* (three species), *Pseudozyma* (two species) and *Ustilago* (one species). In addition, three strains were found to be two potential new species in genus *Rhodotorula* (two strains) and *Vishniacozyma* (one strain). Among the species obtained, strains of *Moesziomyces antarcticus* were found in 55 rice leaf samples (61.8%) followed by strains of *Dirkmeia churashimaensis*, found in 36 rice leaf samples (40.4%) (Table 3).

### 3.2. Selection of Antagonistic Yeasts Capable of Antagonizing Fungi Causing Rice Diseases

Antagonistic activity of rice phylloplane yeast strains against rice pathogenic fungi, namely, *F. moniliforme DOAC 1224*, *H. oryzae DOAC 2293*, *R. solani DOAC 1406*, *Cu. lunata DOAC 2313* and *P. oryzae*, were investigated. In this study we used 80 rice phylloplane yeast strains, consisting of 26 strains in eight species in Ascomycota and 54 strains in 18 species in Basidiomycota (Table 3). The results showed that only 14 yeast strains of five species (*Kodamaea ohmeri*, *M. caribbica*, *M. guilliermondii*, *Torulaspora indica* and *W. anomalus*) belonging to the phylum Ascomycota could inhibit growth of rice pathogenic fungi; none of the strains in the phylum Basidiomycota inhibited any of the species of rice pathogenic fungi. Among antagonistic strains, three yeast strains, namely, *T. indica* DMKU-RP31, *T. indica* DMKU-RP35 and *W. anomalus* DMKU-RP25, inhibited all five rice pathogenic fungal species. The other strains inhibit one to three species of rice pathogenic fungi (Table 4). Among these antagonistic yeast strains, two strains of *T. indica* (DMKU-RP31 and DMKU-RP35) showed the strongest inhibition against all rice pathogenic fungal species except *Cu. lunata* DOAC 2313, whereby these two yeast strains showed equal inhibition to that of *K. ohmeri* DMKU-RP233.

### 3.3. Antagonistic Mechanisms of Antagonistic Yeasts

#### 3.3.1. Production of Antifungal Volatile Organic Compounds

The 14 antagonistic yeast strains were tested for their production of antifungal VOCs capable of pathogenic fungal growth inhibition using double Petri dish assays. The results revealed that most of the strains produced VOCs inhibiting growth of the rice pathogenic fungi (Table 5). Among the strains tested, *T. indica* DMKU-RP31 and *T. indica* DMKU-RP35 were capable of producing VOCs that showed the strongest growth inhibition of *Cu. lunata* DOAC 2313 (60.2 and 59.6%), *F. moniliforme* DOAC 1224 (50.9 and 51.2%), *P. oryzae* (91.9%) and *R. solani* DOAC 1406 (94.1%), whereas VOCs produced by *W. anomalus* DMKU-RP25 also inhibited growth of *R. solani* DOAC 1406 by 94.1%. In addition, VOCs produced by *T. indica* DMKU-RP31 and *W. anomalus* DMKU-RP25 revealed the highest (49.3%) inhibition of *H. oryzae* DOAC 2293 among the five pathogens tested. 

#### 3.3.2. Production of β-Glucanase and Chitinase

Production of β-glucanase and chitinase of 14 antagonistic yeast strains was determined by cultivation in PDB at 25 °C for 5 days. The results revealed that 12 strains produced small amounts (0.2–27.1 mU/mL) of β-glucanase, with *K. ohmeri* DMKU-RP34 producing the greatest amount of β-glucanase (Table 6). Eight antagonistic yeast strains produced small amounts (2.0–249.2 mU/mL) of chitinase; *K. ohmeri* DMKU-RP233 produced the largest amount of chitinase.

#### 3.3.3. Competition for Nutrients and Space

The effect of nutrient concentration on the antagonistic activity of the antagonistic yeast strains against the five stains of rice pathogenic fungi was examined using dual culture on PDA with different nutrient concentrations. The results demonstrated that all the antagonistic strains showed the highest inhibition of the mycelium growth of rice pathogenic fungi when dual cultured on standard PDA with standard nutrient concentration (39 g/L PDA powder), and the reduction of growth inhibition was observed when half of the standard nutrient concentration was used, and most of the antagonistic yeast strains failed to inhibit the pathogenic fungal growth when one-fourth and one-tenth of the standard nutrient concentration were used (Table 5).

#### 3.3.4. Phosphate and Zinc Oxide Solubilization

Phosphate and zinc oxide solubilizing activity of all antagonistic yeasts determined on Pikovskaya’s agar and zinc oxide agar showed that only four strains grew and produced halo zones around colonies. The phosphate and zinc oxide solubilization efficiency (SE) units were calculated to be 1.0–1.2 (Table 6).

#### 3.3.5. Siderophore Production

Determination of the siderophore production of the antagonistic yeast strains on chrome azurol sulfonate (CAS) agar dishes revealed that only two strains of *W. anomalus* (DMKU-RP04 and DMKU-RP25) grew well and formed orange halo zones (2.9–3.0 cm) around the colony (Table 6). This indicated that these two antagonistic yeast strains produced siderophores.

#### 3.3.6. Biofilm Formation

Biofilm formation of the selected antagonistic yeast strains was determined based on absorbance values obtained from the negative control (A_620_ = 0.0745). The results indicated that 13 strains formed biofilms and *M. guilliermondii* DMKU-RP26 did not form biofilm (Table 6).

### 3.4. Controlling of Rice Sheath Blight Disease in Rice Plants in the Greenhouse by the Selected Antagonistic Yeasts

Three antagonistic yeast strains, namely *T. indica* DMKU-RP31, T. indica DMKU-RP35 and *W. anomalus* DMKU-RP25, which inhibited growth of *R. solani* DOAC 1406, the cause of sheath blight disease, by 86.3, 85.4 and 79.7%, respectively, were selected and tested for control of rice sheath blight disease in rice plants in the greenhouse. Forty-five-day-old rice seedlings were sprayed with a cell suspension of one of the three antagonistic yeast strains or a chemical fungicide, 3% validamycin. Fifteen days after pathogenic fungal inoculation, rice sheath blight disease symptoms were observed and lesion height was measured. The result revealed that *T. indica* DMKU-RP31, *T. indica* DMKU-RP35 and *W. anomalus* DMKU-RP25 suppressed sheath blight disease by 70.3%, 66.0% and 66.4%, respectively (Table 7, Figure 1). On the other hand, 3% validamycin suppressed this disease by 83.8%, which was higher than any of the three antagonistic yeast strains.

### 3.5. Yeast Population and Development of Sheath Blight Lesion on Rice Plant

The population of *T. indica* DMKU-RP31 on the rice sheath surface during controlling rice sheath blight in a greenhouse experiment was examined and the results are shown in Figure 2a. The highest population of *T. indica* DMKU-RP31 (7.32x10^4^ CFU/cm^2^) was observed at 0 day after spraying of yeast cell suspension (10^8^ cells/mL) for 1 h, and then the population continuously decreased and reached the lowest (2.58 × 10 CFU/cm^2^) at 5 days. The second and third sprayings of yeast cell suspension (10^8^ cells/mL) resulted in increasing of the yeast population to 7.29 × 10^5^ and 8.74 × 10^5^ CFU/cm^2^, respectively, after spraying for 1 h. However, the population continuously decreased again to 2.51 × 10 and 2.52 × 10 CFU/cm^2^, respectively, at 5 days.

The development of sheath blight lesion when inoculated with *R. solani* DOAC 1406 alone and with *T. indica* DMKU-RP31 was observed by measuring the lesion height. The results showed that the first lesion was observed at 2 days after inoculation of pathogenic fungus and continued to increase in both cases. However, inoculation of the antagonistic yeast with the pathogenic fungus resulted in lower lesion height than when inoculated with the pathogenic fungus alone (Figure 2b).

## 4. Discussion

In this work we isolated rice phylloplane yeasts by plating of leaf washing using YM agar and obtained a higher number of yeast strains in phylum Basidiomycota (84.4%) than in the phylum Ascomycota (15.6%). This result is in accordance with previous reports when the same isolation technique, the plating of leaf washings, was used to isolate yeast from sugarcane phylloplane for a diversity study in Brazil [4] and in Thailand [60]. Both investigations reported a majority of basidiomycetous yeast strains and a smaller number of ascomycetous strains. In the present study, *Moesziomyces antarcticus* was found to be present in as many as 55 rice leaf samples. Our result agrees well with the result of Nasanit et al. [7] that *M. antarcticus* was the most frequently detected species in the rice phylloplane when a culture-independent method was used to assess yeast diversity. Among 50 species obtained from rice phylloplane, ten species (*C. tropicalis*, *D. nepalensis*, *M. antarcticus, M. aphidis P. flavescens*, *P. laurentii*, *P. rajasthanensis*, *R. taiwanensis* and *Sp. blumeae*) were also detected when an enrichment isolation technique was used [61].

Various yeast species in both phyla have been reported to have antagonistic activity against plant pathogenic fungi. Examples of ascomycetous yeast species were *C. oleophila*, *C. sake*, *Hanseniaspora uvarum*, *K. ohmeri*, *Metschnikowia fructicola*, *M. guilliermondii*, *S. cerevisiae* and *Torulaspora globosa*, and examples of basidiomycetous yeast specieswere *Cryptococcus albidus*, *P. laurentii* and *Sp. pararoseus* [21,25,62,63,64,65]. In the present study, only 83 yeast strains out of 282 strains with active growing were evaluated for their antagonistic activities. This was because many yeast strains showed very weak growth or lost their viability after preservation at −80 °C for many months. The results revealed that 14 strains of five yeast species were capable of inhibiting growth of one to five rice pathogenic fungi, namely *Cu. lunata*, *F. moniliforme*, *H. oryzae*, *P. oryzae* and *R. solani.* Among the inhibiting species, *T. indica* and *W. anomalus* inhibited all five rice pathogenic fungi. The results indicated that among microflora associated with rice phylloplane, some antagonistic yeasts that have potential to control rice diseases caused by fungi are present. In this study, only ascomycetous yeast species were capable of antagonizing these fungal pathogens of rice. Some antagonistic yeast species found in the present study have been reported previously. *M. guilliermondii* could control the chilli anthracnose fungus after harvesting [31]. *K. ohmeri* revealed growth inhibition of *Penicillium expansum*, a postharvest pathogenic fungus [62]. *W. anomalus* was reported for its biocontrol activity against *Alternaria alternata*, *Aspergillus carbonarius*, *Botrytis cinerea*, *Monilinia fructicola* and *P. digitatum* [66]. Among *Torulaspora* species, only *T. globose* was previously reported for its biocontrol activity against *Colletotrichum sublineolum* in sorghum [21]. This is the first report on the strong antagonistic activity of *T. indica* strains against pathogenic fungi causing rice diseases, namely *Cu. lunata*, *F. moniliforme*, *H. oryzae*, *P. oryzae* and *R. solani*

Direct and indirect antagonistic mechanisms of the antagonistic yeast strains obtained in this study were evaluated. This study showed that VOC production played a role in the antagonistic activity of *T. indica, W. anomalus* and *K. ohmeri* against *Cu. lunata*, *F. moniliforme*, *P. oryzae* and *R. solani* but not *H. oryzae*. Our findings agree with those reports that emission of VOCs by antagonistic yeasts have proven to be one of the important direct antagonistic mechanisms against pathogenic fungi [27,28,29,30]. In this study, some yeast strains produced the fungal cell wall lytic enzymes β-1, 3-glucanase (12 strains) and chitinase (six strains) in liquid medium, although with relatively low activities. Therefore, the production of β-1,3-glucanase and chitinase seems to be one of the direct antagonistic mechanisms of the antagonistic yeast for inhibition of pathogenic fungi used in this study, which is same as that observed in various antagonistic yeast such as *C. oleophila*, *M. guilliermondii* and *P. membranifaciens* [22,31]. Competition for nutrients between antagonists and pathogenic fungi is among the direct antagonistic mechanisms. This mechanism is not easy to demonstrate on plants because it is difficult to control the other mechanisms [67]. In this study, we tested this mechanism in vitro on PDA with different nutrient concentration. The concentration used in the tests could be different from that available in plants. Unfortunately, we did not check the concentration of nutrients in the rice sheath. The results of the present study showed that the highest fungal mycelial growth inhibition was observed when standard nutrient concentration was used and the efficacy of fugal mycelial growth inhibition decreased when cultured at lower nutrient concentrations. These results could be interpreted to mean that nutrient competition was not a mechanism of these antagonistic yeasts against rice pathogenic fungi. Our study revealed that all of the antagonistic yeasts, except *M. guilliermondii* DMKU-RP26, showed the ability to form biofilms when grown in PDB. This ability could be one of the direct antagonistic mechanisms of these antagonistic yeasts. In the present study, we found that only the strains of *W. anomalus* produced siderophores. Therefore, this could be one of the antagonistic mechanisms by which *W. anomalus* controls the tested rice pathogenic fungi. In this study, strains of *T. indica* and *W. anomalus* showed phosphate and zinc oxide solubilizing activities. Our result appeared to be the same as that of *T. globosa* which exhibited phosphate solubilization to promote plant growth and be one of the biocontrol mechanisms [21]. Our results indicated that for *T. indica* and *W. anomalus*, VOC production was the major mechanism, whereas production of β -1,3-glucanase and chitinase, biofilm formation, and phosphate and zinc oxide solubilization were hypothesized as possible additional mechanisms. On the other hand, in the other yeast species, such as *K. ohmeri, M. caribbica* and *M. guilliermondii*, the antagonistic activity may result from the production of VOCs, β -1,3-glucanase and chitinase and biofilm, however, they did not have the ability to solubilize phosphate and zinc oxide. The antagonistic mechanisms determined in vivo should be further studied.

In this study, the antagonistic yeast cell concentration used in controlling rice sheath blight was 10^8^ cells/mL, which was the same as that used by Rosa et al. [21] to evaluate the efficacy of *T. globosa* in controlling anthracnose in sorghum caused by fungus. They reported that this concentration significantly reduced the anthracnose of sorghum. Recently, Khunamwong et al. [63] reported that it was possible to significantly suppress rice sheath blight disease in a greenhouse by spraying a 10^8^ cells/mL of *W. anomalus* DMKU-CE52 and *W. anomalus* DMKU-RE13. In this study, examination of yeast population on rice sheath surface during controlling rice sheath blight revealed a decrease of population from 7.29–8.74 × 10^4^ CFU/cm^2^ to 2.51–2.58 × 10 CFU/cm^2^ after 5 days of spraying yeast cell suspension. However, the disease lesion development when inoculating *R. solani* with yeast was lower when compared to inoculating *R. solani* alone. This indicated that a yeast population of 10^2^–10^5^ CFU/cm^2^ was enough to reduce sheath blight disease. However, Fokkema et al. [68] suggested that at least 10^4^ CFU/cm^2^ of yeast was necessary to control necrotrophic fungal pathogens on rye and wheat leaves and the higher values of antagonist populations might be required to obtain a better control of decay. The decreasing of yeast population on leaf surface was already found on bean leaves when applied with *R. glutinis* or *Cry. albidus* [69] and on wheat leaves applied with *Sp. roseus* or *Cry. laurentii* [70].

Sheath blight is the second most important rice disease worldwide after blast [17,18]. In this study, strains of *T. indica* (two strains) and *W. anomalus* (one strain) were evaluated for their ability to control rice sheath blight disease in rice plants in the greenhouse. Although the suppression of rice sheath blight by these antagonistic yeast strains was high, it did not reach the level of efficacy of the chemical fungicide, validamycin. However, using antagonistic yeasts as biocontrol agents in agricultural crops is an environmentally friendly alternative method. Some bacteria, actinomycetes and yeast strains have been found to be capable of controlling rice sheath blight disease in the greenhouse. A strain of *Streptomyces philanthi* was found to be effective in the control of rice sheath blight disease in the greenhouse when either spores or a cell suspension was applied [14]. *Sporobolomyces* sp. LR951565 was reported to control rice sheath blight disease in the greenhouse [71]; however, the efficacy of this yeast strain seems to be less than the strains in this study. *W. anomalus* DMKU-CE53 and *W. anomalus* DMKU-RE13 were reported to control rice sheath blight disease in the greenhouse and the biocontrol efficiency were 55.2–65.1%. This is similar to the efficiency of *W. anomalus* DMKU-RP25, the strain in this study (66.4%.) [65]. To our knowledge, this is the first report of using *T. indica* for the biocontrol of rice sheath blight disease caused by *R. solani*.

## Figures and Tables

**Figure 1 microorganisms-08-00362-f001:**
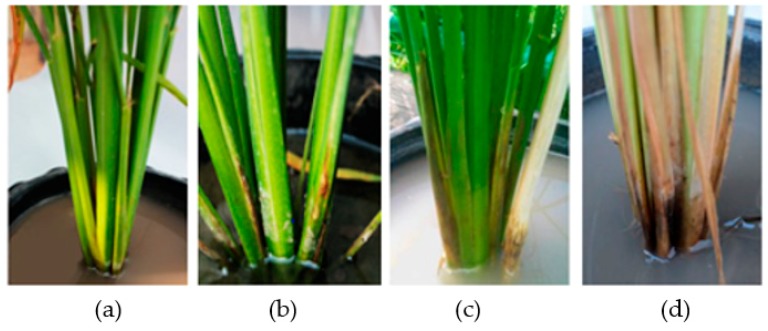
Rice sheath blight disease lesions 15 days after *R. solani* DOAC 1406 inoculation. (**a**) Negative control: rice plant sprayed with distilled water, (**b**) rice plant inoculated with *R. solani* DOAC 1406 and sprayed with 3% validamycin, (**c**) rice plant inoculated with *R. solani* DOAC 1406 and sprayed with cell suspension of *T. indica* DMKU-RP31 and (**d**) positive control: rice plant inoculated with *R. solani* DOAC 1406 without any treatment.

**Figure 2 microorganisms-08-00362-f002:**
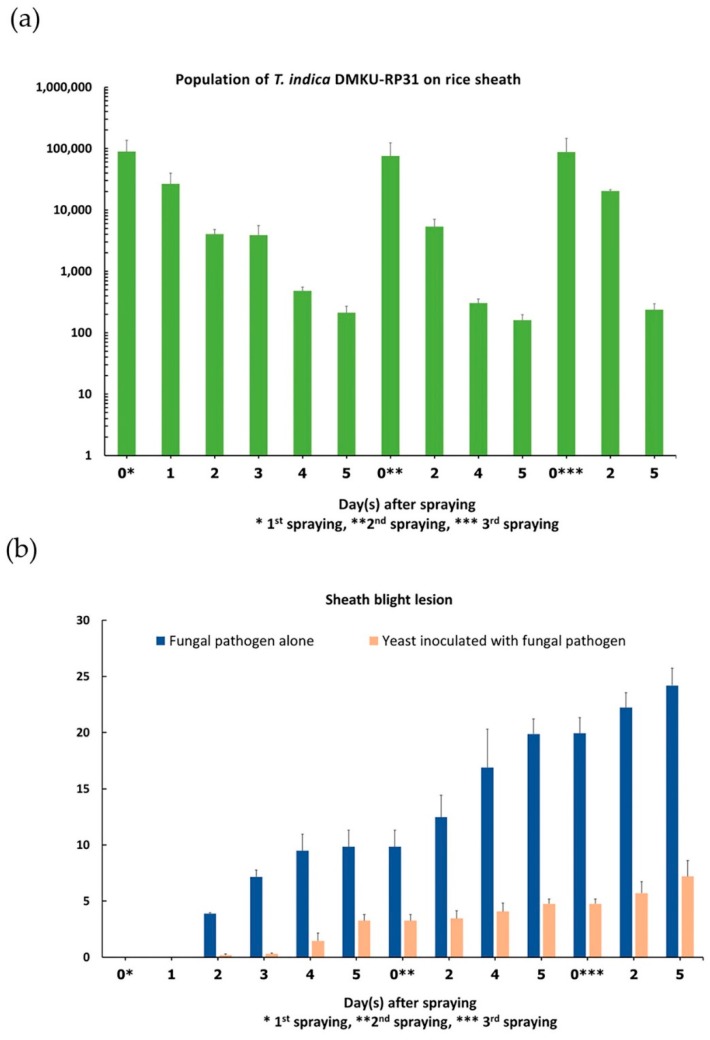
Yeast population and sheath blight diseases development. (**a**) Population of *T. indica* DMKU-RP31, (**b**) rice sheath blight lesion height.

**Table 1 microorganisms-08-00362-t001:** Rice leaf samples collected in Thailand and the number of yeast strain isolated from the phylloplane.

Province	District	Location	Sampling Month and Year	No. of Samples	No. of Strains
Chachoengsao	Ban Pho	13°35′46.0″N 01°04′56.8″E	Dec 2011	1	3
	Bang Pakong	13°29′46.3″N 00°57′14.5″E	Dec 2011	1	1
	Bang Khla	13°41′11.5″N 01°04′13.3″E	Dec 2011	2	8
	Mueang Chachoengsao	13°43′53.0″N 00°59′22.8″E	Dec 2011	1	6
	Phanom Sarakham	13°45′54.1″N 01°19′42.5″E	Dec 2011	2	7
	Ratchasan	13°48′45.8″N 01°16′54.9″E	Dec 2011	1	2
Chai Nat	Manorom	15°20′37.9″N 00°08′52.0″E	Mar 2012	3	7
	Mueang Chai Nat	15°13′28.7″N 00°05′45.7″E	Mar 2012	3	7
Kanchanaburi	Phanom Thuan	14°09′08.7″N 99°40′48.2″E	Jan 2012	4	14
Nakhon Nayok	Ban Na	14°15′34.0″N 01°01′51.4″E	Dec 2011	4	8
	Mueang Nakhon Nayok	14°15′29.3″N 01°13′04.8″E	Dec 2011	1	2
	Pak Phli	14°19′34.6″N 01°21′47.7″E	Feb 2012	4	6
Nakhon Pathom	Bang Len	14°01′55.8″N 00°09′08.6″E	Jan 2012	3	7
	Don Tum	13°58′00.6″N 00°04′13.7″E	Jan 2012	1	3
	Kamphaeng Saen	14°04′53.0″N 99°57′02.3″E	Jan 2012	8	29
Nakhon Sawan	Mueang Nakhon Sawan	15°45′10.4″N 00°07′38.4″E	Mar 2012	2	11
	Phayuha Khiri	15°30′33.9″N 00°09′51.0″E	Mar 2012	1	2
Nonthaburi	Bang Bua Thong	13°55′56.9″N 00°24′37.1″E	Feb 2012	1	10
	Sai Noi	14°01′04.3″N 00°18′55.7″E	Jan 2012	8	36
Prachin Buri	Mueang Prachin Buri	14°08′44.1″N 01°22′55.1″E	Dec 2011	2	6
	Si Mahosot	13°55′01.6″N 01°24′23.2″E	Dec 2011	2	5
Suphan Buri	Bang Pla Ma	14°23′02.7″N 00°08′46.8″E	Mar 2012	7	20
	Doem Bang Nang Buat	14°52′00.9″N 00°09′29.9″E	Mar 2012	1	1
	Don Chedi	14°39′53.3″N 99°57′59.0″E	Jan 2012	3	9
	Mueang Suphan Buri	14°26′57.4″N 00°03′44.0″E	Jan 2012	7	23
	Song Phi Nong	14°13′02.3″N 99°58′44.9″E	Jan 2012	4	12
	U Thong	14°26′27.5″N 99°52′39.5″E	Jan 2012	12	37

**Table 2 microorganisms-08-00362-t002:**
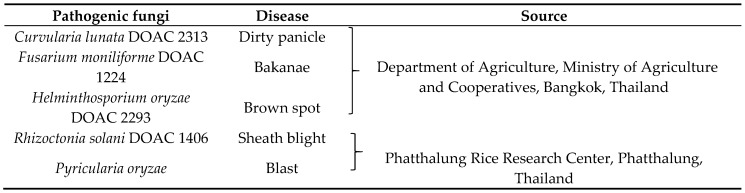
Rice pathogenic fungi used in this study.

**Table 3 microorganisms-08-00362-t003:** Yeast species and strains belonging to the phyla Ascomycota and Basidiomycota in rice phylloplane.

Taxa	No. of Strains	Frequency of Occurrence (%) ^c^	No. of Strains Evaluated for Antagonistic Activity
**Phylum Ascomycota, Subphylum Saccharomycotina**			
*Blastobotrys arbuscular*	1	1.1	
*Candida diddensiae*	1	1.1	
*Candida maltosa*	2	2.2	2
*Candida parapsilosis*	6	6.7	
*Candida tropicalis*	4	4.5	2
*Candida wangnamkhiaoensis*	1	1.1	
*Debaryomyces nepalensis*	1	1.1	1
*Hyphopichia burtonii*	1	1.1	
*Kodamaea ohmeri*	7	7.9	7
*Meyerozyma caribbica*	11	12.4	11
*Meyerozyma guilliermondii*	2	2.2	1
*Torulaspora indica*	2	2.2	2
*Wickerhamomyces anomalus*	3	3.4	2
*Wickerhamomyces edaphicus*	1	1.1	
*Yamadazyma epiphylla* ^a^	1	1.1	
**Phylum Basidiomycota, Subphylum Agaricomycotina**			
*Hannaella sinensis*	4	4.5	
*Hannaella siamensis* ^b^	7	7.9	2
*Hannaella pagnoccae*	1	1.1	1
*Hannaella phetchabunensis*	2	2.2	
*Papiliotrema aspenensis*	4	4.5	
*Papiliotrema flavescens*	7	7.9	
*Papiliotrema japonica*	15	16.9	5
*Papiliotrema laurentii*	2	2.2	
*Papiliotrema nemorosus*	1	1.1	1
*Papiliotrema rajasthanensis*	5	5.6	
*Papiliotrema siamense*	3	3.4	
*Saitozyma flava*	2	2.2	2
*Trichosporon asahii*	1	1.1	
*Trichosporon asteroides*	1	1.1	
*Trichosporon insectorum*	1	1.1	
Potential new species closest to *Vishniacozyma taibaiensis*	1	1.1	
**Phylum Basidiomycota, Subphylum Pucciniomycotina**			
*Occultifur plantarum*	2	2.2	2
*Rhodotorula mucilaginosa*	9	10.1	4
*Rhodotorula paludigena*	7	7.9	1
*Rhodotorula taiwanensis*	24	27.0	12
*Rhodotorula toruloides*	2	2.2	2
Potential new species closest to *Rhodotorula toruloides*	2	2.2	2
*Sakaguchia oryzae*	5	5.6	3
*Sporobolomyces blumeae*	14	15.7	1
*Sporobolomyces carnicolor*	4	4.5	
*Sporobolomyces nakasei*	1	1.1	
*Sporidiobolus pararoseus*	7	7.9	3
*Symmetrospora vermiculata*	2	2.2	1
**Phylum Basidiomycota, Subphylum Ustilaginomycotina**			
*Dirkmeia churashimaensis*	36	40.4	7
*Jaminaea angkoriensis*	3	3.4	
*Kalmanozyma vetiver*	1	1.1	
*Moesziomyces antarcticus*	55	61.8	5
*Moesziomyces aphidis*	1	1.1	
*Moesziomyces parantarcticus*	1	1.1	1
*Pseudozyma alboarmeniaca*	2	2.2	
*Pseudozyma hubeiensis*	1	1.1	
*Ustilago siamensis*	2	2.2	

^a^ Jindamorakot et al. [58]. ^b^ Kaewwichian et al. [59]. ^c^ Frequency of occurrence (%) was calculated as number of samples, where a particular species was observed, as a proportion of the total number of samples.

**Table 4 microorganisms-08-00362-t004:** Growth inhibition of rice pathogenic fungi by yeasts on potato dextrose agar (PDA) at 25 °C for 7 days.

Yeast	Growth Inhibition by Yeast (%) ^a^				
	*Cu. lunata* DOAC 2313	*F. moniliforme* DOAC 1224	*H. oryzae* DOAC 2293	*R. solani* DOAC 1406	*P. oryzae*
*Kodamaea ohmeri*					
DMKU-RP06	48.3 ± 1.4d	23.3 ± 5.73cd	0	0	0
DMKU-RP18	0	20.0 ± 3.57e	0	0	0
DMKU-RP24	0	16.7 ± 4.72f	0	0	45.9 ± 6.5c
DMKU-RP34	0	25.1 ± 2.31c	0	0	0
DMKU-RP44	0	23.2 ± 3.89cd	0	0	0
DMKU-RP57	44.7 ± 1.5e	46.6 ± 2.53a	0	0	0
DMKU-RP233	63.1 ± 0.5a	23.3 ± 4.75cd	0	0	38.8 ± 7.8d
*Meyerozyma caribbica*					
DMKU-RP07	38.0 ± 0.6	23.3 ± 3.27cd	0	0	0
DMKU-RP55	0	25.0 ± 2.9c	59.8 ± 0.86bc	0	33.5 ± 7.9d
*Meyerozyma guilliermondii*					
DMKU-RP26	0	15.2 ± 2.6g	0	0	43.4 ± 8.7c
*Torulaspora indica*					
DMKU-RP31	62.0 ± 1.7ab	46.6 ± 3.2a	64.1 ± 0.7a	86.3 ± 0.9a	62.6 ± 4.4a
DMKU-RP35	61.0 ± 0.9ab	46.6 ± 3.2a	64.9 ± 1.5a	85.4 ± 0.8a	62.2 ± 4.6a
*Wickerhamomyces anomalus*					
DMKU-RP04	50.3 ± 1.7c	30.0 ± 2.0b	48.5 ± 2.9d	0	0
DMKU-RP25	59.1 ± 1.0b	29.9 ± 2.1b	60.2 ± 1.1b	79.7 ± 0.5b	55.7 ± 6.6b

^a^ Inhibition (%) = (radius of control fungal colony-Radius of fungal colony grow with yeast)/radius of control fungal colony × 100. Each value represents a mean “±” standard deviation (SD). In the same column, data followed by different lower-case letters are significantly different according to Duncan’s multiple range test at *p* ≤ 0.05.

**Table 5 microorganisms-08-00362-t005:** Production of antifungal volatile organic compounds and competition of nutrients of the antagonistic yeasts against rice pathogenic fungi.

Rice Pathogenic Fungus and Antagonistic Yeast	Growth Inhibition by VOCs (%) ^a^	Growth Inhibition in Different Nutrient Competition ^b^				
		A ^c^	B ^d^	C ^e^	D ^f^	Sum
***Cu. lunata* DOAC 2313**						
*K. ohmeri* DMKU-RP06	15.2 ± 2.0d	48.3a	18.8b	6.3c	0d	-
*K. ohmeri* DMKU-RP57	35.1 ± 0.5b	44.7a	15.1b	0c	0c	-
*K. ohmeri* DMKU-RP233	19.4 ± 0.3cd	31.4a	7.5b	0c	0c	-
*M. caribbica* DMKU-RP07	32.1 ± 0.6bc	38.0a	17.2b	0c	0c	-
*T. indica* DMKU-RP31	60.2 ± 0.3a	62.1a	38.4b	6.9c	0d	-
*T. indica* DMKU-RP35	59.6 ± 0.9a	61.0a	38.3b	4.4c	0d	-
*W. anomalus* DMKU-RP04	15.7 ± 1.1d	50.3a	22.8b	6.1c	0d	-
*W. anomalus* DMKU-RP25	23.8 ± 1.2c	59.1a	21.1b	6.1c	0d	-
***F. moniliforme*** DOAC 1224						
*K. ohmeri* DMKU-RP06	6.8 ± 1.3e	23.3a	0b	0b	0b	-
*K. ohmeri* DMKU-RP18	25.3 ± 1.2c	20.0a	6.9b	0c	0c	-
*K. ohmeri* DMKU-RP24	17.6 ± 6.9d	16.7a	10.1b	0c	0c	-
*K. ohmeri* DMKU-RP34	20.2 ± 0.5cd	25.1a	8.7b	0c	0c	-
*K. ohmeri* DMKU-RP44	23.3 ± 0.7cd	23.2a	8.6b	2.3c	0d	-
*K. ohmeri* DMKU-RP57	0f	46.6a	25.6b	11.9c	0d	-
*K. ohmeri* DMKU-RP233	22.1 ± 1.1cd	23.3a	5.8b	0c	0c	-
*M. caribbica* DMKU-RP07	6.4 ± 1.9e	23.3a	6.8b	0c	0d	-
*M. caribbica* DMKU-RP55	0f	25.0a	11.6b	1.6c	0d	-
*M. guilliermondii* DMKU-RP26	0f	15.2a	3.8b	0c	0c	-
*T. indica* DMKU-RP31	50.9 ± 0.5a	46.6a	28.1b	11.1c	0d	
*T. indica* DMKU-RP35	51.2 ± 1.2a	46.6a	24.7b	5.9c	0d	-
*W. anomalus* DMKU-RP04	41.1 ± 1.4b	30.0a	10.7b	0c	0c	-
*W. anomalus* DMKU-RP25	35.2 ± 2.5b	30.0a	20.0b	5.6c	0d	-
***H. oryzae* DOAC 2293**						
*M. caribbica* DMKU-RP55	0d	59.8a	44.4b	23.5c	0d	-
*T. indica* DMKU-RP31	49.3 ± 0.5a	64.1a	47.3b	25.8c	0d	-
*T. indica* DMKU-RP35	31.5 ± 0.6b	64.9a	47.2b	24.6c	0d	-
*W. anomalus* DMKU-RP04	21.5 ± 0.4c	48.5a	37.9b	23.8c	0d	-
*W. anomalus* DMKU-RP25	49.3 ± 0.5a	60.2a	46.6b	25.6c	0d	-
***R. solani* DOAC 1406**						
*T. indica* DMKU-RP31	94.1 ± 0.0a	86.3a	80.7b	0c	0c	-
*T. indica* DMKU-RP35	94.1 ± 0.0a	85.4a	80.3b	0c	0c	-
*W. anomalus* DMKU-RP25	94.1 ± 0.0a	79.7a	73.3b	0c	0c	-
***P. oryzae***						
*K. ohmeri* DMKU-RP24	9.8 ± 1.0d	36.0a	10.6b	0c	0c	-
*K. ohmeri* DMKU-RP233	73.3 ± 1.1b	27.8a	13.8b	0c	0c	-
*M. caribbica* DMKU-RP55	8.1 ± 0.4d	21.4a	0b	0b	0b	-
*M. guilliermondii* DMKU-RP26	7.2 ± 0.7d	33.3a	10.3b	0c	0c	-
*T. indica* DMKU-RP31	91.9 ± 0.0a	55.8a	26.7b	0c	0c	-
*T. indica* DMKU-RP35	91.9 ± 0.0a	55.3a	26.7b	0c	0c	-
*W. anomalus* DMKU-RP25	52.2 ± 1.0c	47.7a	17.0b	0c	0c	-

^a^ Inhibition (%) = (diameter of control fungal colony - diameter of fungal colony grow with yeast/ diameter of control fungal colony) × 100; each value represents a mean “±” standard deviation (SD). In the same column, for each rice pathogenic fungus tested, data followed by the different lower-case letters are significantly different according to Duncan’s multiple range test at *p* ≤ 0.05. ^b^ Inhibition (%) = (radius of control fungal colony - radius of fungal colony grow with yeast/radius of control fungal colony) × 100; each value represents a mean “±” standard deviation (SD). In the same row, data followed by the different lower-case letters are significantly different according to Duncan’s multiple range test at *p* ≤ 0.05. ^c^ A: standard nutrient concentration (39 g/L PDA powder). ^d^ B: half of standard nutrient concentration (19.5 g/L PDA powder). ^e^ C: one-fourth of standard nutrient concentration (9.7 g/L PDA powder). ^f^ D: one-tenth of standard nutrient concentration (3.9 g/L PDA powder).

**Table 6 microorganisms-08-00362-t006:** Production of β-glucanase and chitinase, phosphate and zinc oxide solubilization, siderophore production, and biofilm formation of the antagonistic yeasts.

Antagonistic Yeast	Enzyme Activities (mU/mL)		SE ^a^		Siderophore Production ^b^	Biofilm Formation		
	Glucanase	Chitinase	Ca_3_(PO)_4_	ZnO		A ^c^	A value ^d^	Sum ^e^
*K. ohmeri* DMKU-RP06	0	25.1 ± 0.5	0	0	0	0.1664 ± 0.03	2.2	+
*K. ohmeri* DMKU-RP18	0.2 ± 0.0	0	0	0	0	0.4848 ± 0.06	6.5	+
*K. ohmeri* DMKU-RP24	4.6 ± 0.9	0	0	0	0	0.5614 ± 0.07	7.5	+
*K. ohmeri* DMKU-RP34	27.1 ± 2.5	2.0 ± 0.4	0	0	0	0.3535 ± 0.02	4.7	+
*K. ohmeri* DMKU-RP44	0	0	0	0	0	0.4905 ± 0.03	6.6	+
*K. ohmeri* DMKU-RP57	11.0 ± 2.1	249.2 ± 39.6	0	0	0	0.1505 ± 0.02	2.0	+
*K. ohmeri* DMKU-RP233	17.8 ± 2.2	0	0	0	0	0.1188 ± 0.01	1.6	+
*M. caribbica* DMKU-RP07	4.7 ± 0.3	88.4 ± 5.9	0	0	0	0.1616 ± 0.01	2.2	+
*M. caribbica* DMKU-RP55	0.6 ± 0.2	0	0	0	0	0.1351 ± 0.02	1.8	+
*M. guilliermondii* DMKU-RP26	4.6 ± 1.2	0	0	0	0	0.0746 ± 0.07	1.0	-
*T. indica* DMKU-RP31	1.7 ± 0.4	35.2 ± 3.5	1.2	1.2	0	0.5407 ± 0.06	7.3	+
*T. indica* DMKU-RP35	2.2 ± 0.3	166.8 ± 5.7	1.2	1.2	0	0.4252 ± 0.03	5.7	+
*W. anomalus* DMKU-RP04	4.2 ± 0.3	109.8 ± 9.4	1.0	1.0	3.0	0.2967 ± 0.01	4.0	+
*W. anomalus* DMKU-RP25	1.8 ± 0.2	107.5 ± 7.0	1.0	1.2	2.9	0.2121 ± 0.02	2.8	+

^a^ Solubilization efficiency (SE) = diameter of the halo zone (cm)/diameter of the colony (cm); ^b^ diameter of holo zone (cm). ^c^ The absorbance of the biofilm layer stained with violet crystal solution measured at 620 nm (average ± SD). ^d^ Average absorbance of samples as a portion of Ac (control); ^e^ Interpretation of biofilm formation: A ≤ Ac, no biofilm formation; Ac < A, biofilm formation. The optical density cut-off value (Ac) = 0.0745.

**Table 7 microorganisms-08-00362-t007:** Efficacy of the antagonistic yeasts in suppressing of rice sheath blight disease caused by *R. solani* DOAC 1406 in rice plants grown in pots in the greenhouse.

Treatment	Plant Height (cm) ^a^	Lesion Height (cm) ^b^	Disease Incidence (%)	Disease Suppression (%)
Control (negative control)	94.0 ± 2.9a	0d	0d	0
*R. solani* (positive control)	92.0 ± 2.6a	23.8 ± 1.6a	25.9 ± 2.3a	0
*R. solani* + *T. indica* DMKU-RP31	93.0 ± 4.1a	7.2 ± 1.4b	7.7 ± 1.4b	70.3
*R. solani* + *T. indica* DMKU-RP35	93.2 ± 2.3a	8.3 ± 1.5b	8.8 ± 1.4b	66.0
*R. solani* + *W. anomalus* DMKU-RP25	92.6 ± 3.0a	8.1 ± 1.0b	8.7 ± 0.8b	66.4
*R. solani* + 3% Validamycin	93.4 ± 3.5a	3.9 ± 0.5c	4.2 ± 0.6c	83.8

Each value represents a mean “±” standard deviation (SD). In the same column, data followed by the different lower-case letters are significantly different according to Duncan’s multiple range test at p≤0.05. ^a^ Plant height was an average height from five rice plants (measured above ground to the tallest leaf). ^b^ Lesion height was an average lesion height from five rice plants; in each plant the height of all lesions was measured and averaged.

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
