# Peer review of "Yeast Associated with Rice Phylloplane and Their Contribution to Control of Rice Sheath Blight Disease"

_microorganisms, 2020, doi:10.3390/microorganisms8030362_

Round 1

Reviewer 1 Report

The manuscript by Into and collaborators report an interesting study on the potential as biocontrol agents of yeast strains isolated from phylloplane. The study is generally reported in clear terms and with scientific rigor, and the experimental design and approaches are appropriate. However, there are minor changes that I believe would improve the manuscript.

Please write all the genera and species in Italic.

The authors used different ways to calculate the inhibition caused by the tested isolates in different tests. For the direct contact inhibition test (standard conditions and various availability of nutrients), they measured the ratio of the inhibition halo, while for the indirect inhibition (VOC) they measured the diameter. For comparison, it would be better to use the same metrics, either measuring the ratio or the diameter for all tests.

There are some inconsistencies with the number of strains reported. For instance, at L285, the authors report they found 44 Ascomycetes belonging to 13 known and 1 new species. However, in Table 3 and in lines 285-289, they reported 15 Ascomycete species. Please correct.

In addition, in line 294, 34 known species are reported, while afterward the authors refer to 36 species.

Furthermore, it is not clear why the authors consider some of the species as “new” even if they were already described in previous studies.

L 17: I guess the authors meant “… and could be used as biocontrol agents.”

L20: please change to “The majority of rice phylloplane yeasts belonged to the Basidiomycota phylum” or “The majority of rice phylloplane yeasts were Basidiomycetes”.

L21: please delete the “:”

L21: were the 80 strains tested yeasts? Or were they fungi? Considering that, as stated, the majority of isolates were basidiomycetes, I would expect the authors tested some of them? Or did they arbitrarily decided to test yeasts only? Fair decision, but should say it.

L29-30: the sentence is not clear, please rephrase

L35: please change to: “The term phylloplane to the parts of plants above the ground and dominated by...”

L39: please correct to “has been reported”, please also consider that it would be less incorrect (I don’t dare to say this way is correct) to say that Ascomycota are Basidiomycota are fungi, not yeast

L41: please correct to “food” (not “foods”)

L90: please correct to “yeasts were identified...” and “...the large subunit...”

L117: please provide the IDs of the submitted sequences

L160 and L169: please change “absorbent” with “absorption”

The header of table 3: please correct to “No. of strains”, and “No. of strains evaluated for antagonistic activity”

L316: please add “)” after anomalus

L323: please correct to “… against which these two yeasts...”

Table 5: It would be useful to show the data reported in this table in different tables. I would report the growth inhibition by VOCs and growth inhibition in different nutrient competition in one table and Glucanase and chitinase activities, biofilm formation, siderophore production, and solubilization efficiency in a different table. In fact, the measures of the latter group are independent of pathogenic fungi (measured in their absence), and hence there is one value for each isolate. Contrarily, the inhibition measures are based on the pathogens, and in fact, differ when a given isolate is challenged against different pathogens.

L361-362. The sentence is not clear, please rephrase

L367 and everywhere there is a reference to the nutritional content assay: it would be more appropriate to refer to “standard” conditions rather than to “normal” conditions.

L368: please change to “.. and ta reduction of growth inhibition...”

It would be important to discuss the effect of the tested isolates alone on the plant. Did the author measure this effect?

L387: please rephrase the sentence. In its current form, the sentence seems to indicate that the isolated strains were inoculated together with the fungicide, while to my understanding the fungicide was tested alone, as the control.

L413: please change to “...then the population continuously decreased...”

L414: It is not clear what the authors meant here. In addition, it is odd that the number of yeast cells measured after the second and third inoculum does not change, considering that before the new inoculum some cells from the previous inoculum were still present (e.g. ~200 CFU-cm2 before the second inoculum). This may reflect some technical issues with the approach and should be discussed. However, authors should test whether, on the same days after each inoculum, the CFU/cm2 differs in subsequent inoculations.

L415: please change to “...the population continuously decreased again.”

L418: the authors report here that the first lesion was observed 3 days after the inoculation of the pathogenic fungus, while the plot in Figure 2b seems to indicate that the lesion appears on the second day. Please check which one is the correct one, or explain better.

L418: please change to “...and continued increase in both cases”

L419: please change lesser to lower

L431: please change to “...was used to isolate yeasts from sugarcane...”

L447: please change to “Among the inhibiting species, T. indica...”

L474: please change to “...and this directly suppresses activities and/or induces death”

L483-484: the sentence is not clear. Do the authors mean that the measurement of low enzymatic activities can be ascribed to the fact that fungal cell wall was not added in the test, and, considering that the fungal cell wall can stimulate the production of the enzymes, the test may be misleading?

L484-486: the sentence is not clear. Please rephrase

L490: please change “normal” to “standard”. In addition, the authors should discuss how the nutrient concentrations tested in this assay are compared to the nutrients available in plants. Is the amount and type of nutrients in the standard condition similar to those on the plant?

L491: please change to “...growth inhibition decreased when...”

L504: please change to “Rhodotorula glutinis has been shown to produce...”

L505: please change to “which suppresses various...”

L509: please change to “..W. anomalus controls the tested rice pathogenic fungi”

L514-519: it is not clear what is different between the effects and proposed mechanisms of the two groups of isolates. Please rephrase

Author Response

Response to the reviewers #1’ comments

Manuscript ID: microorganisms-723029:

Yeast Associated with Rice Phylloplane and Their Contribution to Control of Rice Sheath Blight Disease

The manuscript by Into and collaborators report an interesting study on the potential as biocontrol agents of yeast strains isolated from phylloplane. The study is generally reported in clear terms and with scientific rigor, and the experimental design and approaches are appropriate. However, there are minor changes that I believe would improve the manuscript.

Point 1: Please write all the genera and species in Italic.

Response 1: Corrected all. 

Point 2: The authors used different ways to calculate the inhibition caused by the tested isolates in different tests. For the direct contact inhibition test (standard conditions and various availability of nutrients), they measured the ratio of the inhibition halo, while for the indirect inhibition (VOC) they measured the diameter. For comparison, it would be better to use the same metrics, either measuring the ratio or the diameter for all tests.

Response 2: Both tests are based on fungal growth reduction. The different is only for direct contact inhibition test we measured the radius of fungal colony face to the streak of yeast because the growth at another side of fungal colony is sometimes limited at the edge of dish, while for the indirect inhibition (VOCs) we measured the diameter of fungal colony because yeast was inoculated on different dish bottoms.

Point 3: There are some inconsistencies with the number of strains reported. For instance, at L285, the authors report they found 44 Ascomycetes belonging to 13 known and 1 new species. However, in Table 3 and in lines 285-289, they reported 15 Ascomycete species. Please correct.

In addition, in line 294, 34 known species are reported, while afterward the authors refer to 36 species.

Response 3: We checked and corrected all errors.

Point 4: Furthermore, it is not clear why the authors consider some of the species as “new” even if they were already described in previous studies.

Response 4: We changed new species which already described to be known species and changed the number of known species.

Point 5: L 17: I guess the authors meant “… and could be used as biocontrol agents.”
Response 5: We changed to “… and could be used as biocontrol agents.”.

Point 6: L20: please change to “The majority of rice phylloplane yeasts belonged to the Basidiomycota phylum” or “The majority of rice phylloplane yeasts were Basidiomycetes”.

Response 6: Changed to “The majority of rice phylloplane yeasts belonged to the phylum Basidiomycota”

Point 7: L21: please delete the “:”

Response 7: .Deleted

Point 8: L21: were the 80 strains tested yeasts? Or were they fungi? Considering that, as stated, the majority of isolates were basidiomycetes, I would expect the authors tested some of them? Or did they arbitrarily decided to test yeasts only? Fair decision, but should say it.

Response 8: We evaluated 83 yeast strains for their antagonistic activities because we selected only yeast strains (not yeast-like fungi) with active growing. There were many yeast strains showed very weak growing or lost their viability during storage. We added “In the present study, only 83 yeast strains out of 282 strains with active growing were evaluated for their antagonistic activities. This was because many yeast strains showed very weak growth or lost their viability after preservation at -80°C for many months.”  in L 501-504 of the revised MS.

Point 9: L29-30: the sentence is not clear, please rephrase

Response 9: We changed the sentence before this sentence to “These yeast strains were evaluated for controlling rice sheath blight caused by R. solani in rice plants in the greenhouse and were found to suppress the disease by 60.0-70.3%, whereas 3% validamycin suppressed by 83.8%.” and changed this sentence to “Therefore, they have potential for developing to be used as biocontrol agents for rice sheath blight.” (L28-31).

Point 10: L35: please change to: “The term phylloplane to the parts of plants above the ground and dominated by...”

Response 10: Changed.

Point 11: L39: please correct to “has been reported”, please also consider that it would be less incorrect (I don’t dare to say this way is correct) to say that Ascomycota are Basidiomycota are fungi, not yeast

Response 11: Corrected.

Point 12: L41: please correct to “food” (not “foods”)

Response 12: Corrected.

Point 13: L90: please correct to “yeasts were identified...” and “...the large subunit...”

Response 13: Corrected.

Point 14: L117: please provide the IDs of the submitted sequences

Response 14: Sequence accession numbers were shown in Table S1.

Point 15: L160 and L169: please change “absorbent” with “absorption”

Response 15: Changed to “absorption”.

Point 16: The header of table 3: please correct to “No. of strains”, and “No. of strains evaluated for antagonistic activity”

Response 16: Corrected.

Point 17: L316: please add “)” after anomalus

Response 17: Added.

Point 18: L323: please correct to “… against which these two yeasts...”

Response 18: Corrected to “showed the strongest inhibition against all rice pathogenic fungal species except Cu. lunata DOAC 2313, which these two yeast strains showed equal inhibition to that of..”.

Point 19: Table 5: It would be useful to show the data reported in this table in different tables. I would report the growth inhibition by VOCs and growth inhibition in different nutrient competition in one table and Glucanase and chitinase activities, biofilm formation, siderophore production, and solubilization efficiency in a different table. In fact, the measures of the latter group are independent of pathogenic fungi (measured in their absence), and hence there is one value for each isolate. Contrarily, the inhibition measures are based on the pathogens, and in fact, differ when a given isolate is challenged against different pathogens.

Response 19: Thank you for your comment. We separated Table 5 in to 2 tables, Table 5 and Table 6 as commented.

Point 20: L361-362. The sentence is not clear, please rephrase

Response 20: Rephrased to “This indicated that these two antagonistic yeast strains produced siderophores.”.

Point 21: L367 and everywhere there is a reference to the nutritional content assay: it would be more appropriate to refer to “standard” conditions rather than to “normal” conditions.

Response 21: All “normal”words were replaced by “standard”.

Point 22: L368: please change to “.. and ta reduction of growth inhibition...”

It would be important to discuss the effect of the tested isolates alone on the plant. Did the author measure this effect?

Response 22: Changed to “.. and a reduction of growth inhibition...”.

Point 23: L387: please rephrase the sentence. In its current form, the sentence seems to indicate that the isolated strains were inoculated together with the fungicide, while to my understanding the fungicide was tested alone, as the control.

Response 23: Changed to “...or a chemical fungicide, 3% validamycin”.

Point 24: L413: please change to “...then the population continuously decreased...”

Response 24: Changed to “...then the population continuously decreased...”.

Point 25: L414: It is not clear what the authors meant here. In addition, it is odd that the number of yeast cells measured after the second and third inoculum does not change, considering that before the new inoculum some cells from the previous inoculum were still present (e.g. ~200 CFU-cm2 before the second inoculum). This may reflect some technical issues with the approach and should be discussed. However, authors should test whether, on the same days after each inoculum, the CFU/cm2 differs in subsequent inoculations.
Response 25:  Yeast enumeration was performed after the second and third sprayings of yeast cell suspension for 1 h of the same day. We added clearer information in the M and M (L 306-308).  Also changed the unclear sentence in the Results to “The second and third sprayings of yeast cell suspension (108 cells/mL) resulted in increasing of yeast population to 7.29x105 and 8.74x105 CFU/cm2, respectively,  after of spraying for 1 hour. However, the population continuously decreased again to 2.51 x10 and 2.52 x10 CFU/cm2, respectively, at 5 days.” L-469-472.

We added some discussion as “The decreasing of yeast population on leaf surface was already found on bean leaves when applied with R. glutinis or Cry. albidus [Elad et al. 1994] and on wheat leaves applied with Sp. roseus or Cry. laurentii [Dik et al. 1992].”(L 564 -565).  

Point 26: L415: please change to “...the population continuously decreased again.”

Response 26: Changed.

Point 27: L418: the authors report here that the first lesion was observed 3 days after the inoculation of the pathogenic fungus, while the plot in Figure 2b seems to indicate that the lesion appears on the second day. Please check which one is the correct one, or explain better.

Response 27: Corrected to “first lesion was observed 2 days after the inoculation..”

Point 28: L418: please change to “...and continued increase in both cases”

Response 28: Changed.

Point 29: L419: please change lesser to lower

Response 29: Changed.

Point 30: L431: please change to “...was used to isolate yeasts from sugarcane...”

Response 30: Changed.

Point 31: L447: please change to “Among the inhibiting species, T. indica...”

Response 31: Changed.

Point 32: L474: please change to “...and this directly suppresses activities and/or induces death”

Response 32: Changed.

Point 33: L483-484: the sentence is not clear. Do the authors mean that the measurement of low enzymatic activities can be ascribed to the fact that fungal cell wall was not added in the test, and, considering that the fungal cell wall can stimulate the production of the enzymes, the test may be misleading?

Response 33: We deleted, which may result from in the enzyme production test, we did not add fungal cell wall to the cultivation medium”.

Point 34: L484-486: the sentence is not clear. Please rephrase

Response 34:  We rephrased to “Therefore, the production of β-1,3-glucanase and chitinase seem to be one of the direct antagonistic mechanisms of the antagonistic yeast..”.

Point 35: L490: please change “normal” to “standard”. In addition, the authors should discuss how the nutrient concentrations tested in this assay are compared to the nutrients available in plants. Is the amount and type of nutrients in the standard condition similar to those on the plant?

Response 35: Changed “normal” to “standard”. In this study, we used potato dextrose agar (PDA), we did not compared to the nutrients available in plants, and as your comment we added discussion about this as “Competition for nutrients between antagonists and pathogenic fungi is among the direct antagonistic mechanisms. This mechanism is not easy to demonstrate on plants because it is difficult to control the other mechanisms [Sperandio et al. 2015]. In this study, we tested this mechanism in vitro on PDA with different nutrient concentration. The concentration used in the tests could be different from that available in plants. However, we did not check concentration of nutrients in the rice sheath.” in L 528-532 of the revised MS.

Point 36: L491: please change to “...growth inhibition decreased when...”

Response 36: Changed.

Point 37: L504: please change to “Rhodotorula glutinis has been shown to produce...”

Response 37: Changed.

Point 38: L505: please change to “which suppresses various...”

Response 38: Changed.

Point 39: L509: please change to “..W. anomalus controls the tested rice pathogenic fungi”

Response 39: Changed.

Point 40: L514-519: it is not clear what is different between the effects and proposed mechanisms of the two groups of isolates. Please rephrase

Response 40: Rephrased to “On the other hand, in the other yeast species, K. ohmeri, M. caribbica and M. guilliermondii, the antagonistic activity may resulted from the production of VOCs, β -1,3-glucanase and chitinase and biofilm, whereas, they did not have ability to solubilize phosphate and zinc oxide.

Reviewer 2 Report

The main aim of the work is an ecological study aimed at identifying yeast species associated with rice phylloplane, with potential antifungal activity. I do not know how big could be the interest of the readers in this type work, even because the presentation it is not appealing. For example, all the parts of the discussion highlighting the importance some yeast characteristics as mean of pathogens control (i.e. biofilm forming ability, siderophores, ecc..) should be better explained in the introductory part, to let reader understand the reason of the analysis you perfomed. If you manage to present your analysis and your results in a more attractive way, maybe the value and the consideration of a potential reader could be higher.

There also formatting problems: the style of the text is not uniform, it needs to be amended. There also several strain names not in Italic, in a species description paper this mistake should not be present.

So I invite you to revise this paper in order to make it more clear and easy to follow.

Author Response

Response to the reviewers #1’ comments

Manuscript ID: microorganisms-723029:

Yeast Associated with Rice Phylloplane and Their Contribution to Control of Rice Sheath Blight Disease

Point 1: The main aim of the work is an ecological study aimed at identifying yeast species associated with rice phylloplane, with potential antifungal activity. I do not know how big could be the interest of the readers in this type work, even because the presentation it is not appealing. For example, all the parts of the discussion highlighting the importance some yeast characteristics as mean of pathogens control (i.e. biofilm forming ability, siderophores, ecc..) should be better explained in the introductory part, to let reader understand the reason of the analysis you perfomed. If you manage to present your analysis and your results in a more attractive way, maybe the value and the consideration of a potential reader could be higher.

There also formatting problems: the style of the text is not uniform, it needs to be amended. There also several strain names not in Italic, in a species description paper this mistake should not be present.

So I invite you to revise this paper in order to make it more clear and easy to follow.

Response 1: We moved the part of yeast characteristics as mean of pathogens control to the in introductory part together with some addition information (L69-105).

Round 2

Reviewer 2 Report

The manuscript can be accepted.